# Benign COVID-19 in an Aggressive Case of Aquaporin-4 Neuromyelitis Optica Treated with Tocilizumab

Domizia Vecchio [1,*] , Claudio Solaro [2] , Eleonora Virgilio [1] , Paola Naldi [1], Rugiada Bottero [3], Fabio Masuccio [2] , Marco Capobianco [3] and Roberto Cantello [1]

1   Department of Translational Medicine, Section of Neurology, University of Eastern Piedmont, 28100 Novara, Italy
2   Department of Rehabilitation, C.R.R.F. "Mons. L. Novarese", Loc. Trompone, 13040 Moncrivello, Italy
3   Department of Neurology and Regional Referral Multiple Sclerosis Centre, University-Hospital S. Luigi Gonzaga, 10043 Orbassano, Italy
*   Correspondence: domizia.vecchio@gmail.com; Tel.: +39-0321-373-3964

**Abstract:** Aggressive neuromyelitis optica spectrum disorders (NMOSDs) with antibodies (Abs) against aquaporin-4 (AQP4) can be treated by blocking the interleukin 6 (IL6) pathways with tocilizumab. This IL6-inhibitor was employed to treat coronavirus disease 2019 (COVID-19) pneumonia with unconclusive results. We present a 52-year-old woman with AQP4 NMOSD, unresponsive to rituximab, that stabilized on tocilizumab one year after the disease onset. She was bed-bound and progressively recovered her mobility. During intensive rehabilitation, she presented fever and cough for one week with nasopharyngeal swabs positive for severe acute respiratory syndrome coronavirus 2 (SARS-CoV-2). This mild COVID-19 recovered spontaneously without sequelae, and the monthly tocilizumab infusions were continued for another 10 months. Subsequently, serious and prolonged respiratory and urinary infections caused treatment interruption, and then her disease re-activated. In our case, tocilizumab was effective in preventing NMOSD relapse and was safe to use during SARS-CoV-2 infection.

**Keywords:** SARS-CoV-2; COVID-19; neuromyelitis optica; tocilizumab; aquaporin-4

## 1. Introduction

Neuromyelitis optica spectrum disorders (NMOSDs) with antibodies (Abs) against aquaporin-4 (AQP4) cause optic neuritis, area postrema syndrome and severe spinal attacks that could have poor recovery. Relapse prevention is mandatory to avoid disability [1], and the first choices are immunosuppressants and anti-CD20 monoclonal Abs. An inhibitor of the complement protein C5 (eculizumab) has recently been approved [2].

If unresponsive to rituximab, and if eculizumab IS contraindicated (OR before its approval, as in this case), "aggressive AQP4 NMOSD" inflammatory activity could be reduced by blocking the interleukin 6 (IL6) pathways WITH tocilizumab [2]. This latter IL6 inhibitor has been studied in severe acute respiratory syndrome coronavirus 2 (SARS-CoV-2) infection according to its effect on the life-threatening cytokine release syndrome, with inconclusive evidence of its benefit [3]. COVID-19 outcome in NMOSDs is still debated, with much higher odds of hospitalization and intensive care unit admission than the general population related to age, disability, disease duration [4], comorbidity, and treatment with rituximab [5].

Here, we discuss an example of a patient with aggressive AQP4 NMOSD developing benign coronavirus disease 2019 (COVID-19) during the tocilizumab regimen, despite being severely disabled and having had previous treatment with rituximab. Unfortunately, other ongoing severe respiratory and urinary infections required tocilizumab interruption.

## 2. Case Presentation Section

A 52-year-old female Italian patient was diagnosed with NMOSD in January 2020, when AQP4 Abs were tested after longitudinal extensive transverse myelitis (LETM). Four months before, she presented with an acute D2–D4 inflammatory lesion and recovered with intravenous (IV) steroids. Brain magnetic resonance imaging (MRI) showed few millimetric hyperintensities in the deep white matter of T2-weighted sequences. At first, LETM was related to possible systemic lupus erythematosus according to: a previous history of undetermined arthritis, the presence of antinuclear Abs at low titer (1:160; cut off < 1:160), and anti-double-stranded DNA (32.4 UI/mL; cut off < 30) with normal complement levels and absent antiphospholipid Abs. Consequently, she was first treated with two cycles of cyclophosphamide, causing a moderate increase in liver function. This asymptomatic adverse event resolved in 2 months after stopping the immunosuppressant infusion.

In the meantime, she had a cervical relapse, causing asymmetric tetraparesis (severe on the right side) and a C5 sensory level. She was treated with IV steroids and plasma exchange, and partially recovered her upper limb function (mainly on the left side). At that time, AQP4 Abs were found to be positive. She started rituximab (standard regimen of two doses, 2 weeks apart, of 1 g each, and subsequent cycles every 6 months). After the third administration, in October 2020, she presented a cervico-bulbar relapse, becoming tetraplegic with vertigo, nausea and nystagmus. B cells were absent at that time. She was immediately treated with IV steroids, plasma exchange, and IV immunoglobulins, and then, in November 2020, tocilizumab was started (8 mg/kg every 28 days). At the time of the second infusion, the patient was bed-bound on an intensive rehabilitation program.

Ten days later, she presented fever and cough with no respiratory distress. She underwent: a nasopharyngeal swab for SARS-CoV-2 (positive on day 1), arterial blood gas (normal with no oxygen support), chest X-ray (no signs of pneumonia), and routine blood tests (normal leukocyte count with absent B lymphocyte, normal C-reactive protein level). No medication was added, and she recovered in one week. A second swab remained positive on day 10 and only a third negativized on day 14. This mild COVID-19 infection resolved spontaneously without sequelae. On day 15 (January 2021) she was given her monthly tocilizumab infusion with a delay of 6 days. Over the months, she progressively recovered and became able to stay in a wheelchair, to use her arms, and to be active all day.

She remained on regular tocilizumab infusions for another 10 months. Subsequently, consecutive serious and prolonged respiratory and urinary infections caused treatment interruption. First, in December 2021, she had an episode of desaturation with fever that required hospitalization. A chest X-ray showed lung atelectasis that complicated with pneumonia, requiring non-invasive ventilation and IV antibiotics. Absolute neutrophil count was within the normal range at that time. A few weeks later, after being discharged, she developed a bacterial urinary infection with sepsis that required another course of IV antibiotics. Still, the routine blood tests evidenced no neutropenia. In the meantime, in March 2022, she presented with a loss of vision in the right eye. In the suspicion of an optic relapse, we commenced a course of IV immunoglobulins with poor effect (steroids and plasma exchange were discussed and avoided because of the recent ongoing infections). Once she had recovered from the cystitis, we decided to start an oral immunosuppression as a maintenance therapy. An amount of 75 mg azathioprine daily was gradually commenced and stopped after one month due to a moderate increase in liver function. We then switched to 7.5 mg methotrexate weekly, and since then, the patient has been clinically stable. Neuropathic pain and spasticity-related symptoms have reduced over time, and she is actually able to stay in a wheelchair 12 h a day. A spinal MRI showed C1-D10 hyperintensity in the T2-weighted sequences with no enhancement.

## 3. Discussion

We have described a severely disabled woman with an aggressive AQP4 NMOSD that was unresponsive to rituximab. In fact, one year after the disease onset, she had severe tetraparesis related to several cervico-dorsal attacks. Only a third-line treatment

with tocilizumab stopped her relapses. During this regimen, she developed a mild SARS-CoV-2 infection, despite being almost bed-bound at that time. She required no oxygen and recovered completely with no additional treatment. Moreover, COVID-19 did not significantly affect her neurological status or her rehabilitation program. Her infusions were delayed until she had a negative swab to avoid transportation during the infection.

COVID-19 prognosis for NMOSD patients ranges from mild disease to severe pneumonia related to age, disability, disease duration [4], comorbidity [5,6], and treatment with rituximab [5]. These risks factors were not confirmed by other authors [7].

Sharifian-Dorche et al. collected 5 severe cases among a cohort of 37 NMOSD patients with COVID-19 that died or were admitted to intensive care units. All of them had high motor disability; 4/5 were on rituximab and 1/5 on steroids [8]. Of note, our patient had her last rituximab infusion two months before the SARS-CoV-2 infection, and B cells were absent when she developed fever related to the infection. Anti-CD20 therapy has been related to a higher risk of COVID-19 [9], mostly if the treatment is recent, as in our case, since a complete B depletion could be observed after a single dose [10]. Moreover, rituximab-treated patients were found to be at a higher risk of developing severe forms of SARS-CoV-2 infection [5].

There are two other cases of mild COVID-19 during tocilizumab treatment for NMOSDs. Mantero et al. presented a 44-year-old medical doctor with AQP4 NMOSD and myasthenia gravis who had a two-day history of fever and abdominal pain with subsequent evidence of IgG and IgM against SARS-CoV-2. She was fully functioning before the infection, and remained so after [11]. Masuccio et al. described a 31-year-old female with MOG NMOSD who had a three-day history of anosmia and generalized myalgia with positive nasopharyngeal swabs. COVID-19 did not impact on her walking abilities, considering she had moderate previous paraparesis [12]. Both previously reported MNO cases were younger and less disabled compared with the patient of our report, but globally all three of them had a mild SARS-CoV-2 infection. No other cases of tocilizumab-treated NMOSD have been reported so far [5].

Moreover, the role of IL6 in COVID-19 cytokine release syndrome has recently been debated, and the benefit of tocilizumab in SARS-CoV-2 infection largely discussed [3]. Nevertheless, we could speculate if the concomitant use of an IL6 inhibitor could facilitate a milder respiratory disease in patients affected by neurological immune-mediated disorders, and add information to the safety issues of NMOSD.

On the other hand, in our case, the occurrence of serious and prolonged respiratory and urinary infections complicated the tocilizumab regimen, causing treatment interruption. The safety profile of this IL6-inhibitor includes, obviously, bacterial infections. However, the risk seems higher in treating rheumatoid arthritis than NMOSD [13], and lower compared to that of azathioprine in neurological clinical trials [14].

Tocilizumab has recently been shown to be safe and effective for preventing relapses in myelin oligodendrocyte glycoprotein IgG-associated disease and seronegative NMOSD [15]. Actually, no guidelines are available for treatment discontinuation. Data on rheumatoid arthritis suggest continuing low-effective drugs (i.e., methotrexate) to maintain disease activity control [16].

Since this is a single case, further evidence is needed to confirm our therapeutic setting. Moreover, eculizumab was not approved yet at the time of this case.

## 4. Conclusions

In our case, tocilizumab was effective in preventing NMOSD relapse as a third-line treatment, and was found to be safe despite COVID-19 infection in a disabled patient.

**Author Contributions:** Conceptualization, D.V. and C.S.; writing—original draft preparation, D.V; writing—review and editing, D.V. and E.V.; supervision, M.C. and R.C.; patient care, P.N., R.B. and F.M. All authors have read and agreed to the published version of the manuscript.

**Funding:** This research received no external funding.

**Institutional Review Board Statement:** Not applicable.

**Informed Consent Statement:** Informed consent has been obtained from the patient to publish this paper.

**Data Availability Statement:** Not applicable.

**Acknowledgments:** Graphical abstract created with BioRender.com (accessed on 9 August 2022).

**Conflicts of Interest:** The authors declare no conflict of interest.

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
