# Peer review of "Benign COVID-19 in an Aggressive Case of Aquaporin-4 Neuromyelitis Optica Treated with Tocilizumab"

_2813-3064, doi:10.3390/sclerosis1010001_

Round 1
Reviewer 1 Report
In the present article, the authors describe a case report of benign COVID19 in a patient with aquaporin-4 neuro-2 myelitis optica treated with tocilizumab. Although the description is interesting to know the relationship between NMO and COVID19 in tocilizumab-treated patients, the manuscript should be improved to be published in this journal.
- The main weakness of the study is the rationale of the case report.
- A better background on the risk of severe COVID19 in patients with NMO should be highlighted.
- The relationship between the characteristics of COVID-19 infection in patients with NMO should be improved.
- What is the importance of previous rituximab treatment in the outcome of COVID19?
- What is the limitation of the case?
- The title is confusing. It should focus on the main message of the manuscript.
Reviewer 2 Report
Introduction:
-- NMO should be replaced with NMOSD
-- the disease does not only cause spinal attacks but also ON and APS, this needs to be amended
-- first choices rituximab: not entirely correct any more as there are newly approved drugs, please correct
- Case presentation:
-- which assay was used to test for AQP abs?
-- was she sufficiently B cell depleted when experiencing the relapse while on RTX?
-- please report results of brain MRI and follow-up spinal cord MRI
- Discussion:
-- some of the topical literature on Covid 19 and NMOSD needs to be discussed, see for example https://pubmed.ncbi.nlm.nih.gov/34446434/
https://pubmed.ncbi.nlm.nih.gov/34429342/
Author Response
Thank you for the suggestions. Changes are highlighted in blue. Here a point-to-point reply.
-- NMO should be replaced with NMOSD: yes, done
-- the disease does not only cause spinal attacks but also ON and APS, this needs to be amended: yes, done
-- first choices rituximab: not entirely correct any more as there are newly approved drugs, please correct: eculizumab was not approved yet at the time of this case, but added in the introduction and as a limit.
- Case presentation:
-- which assay was used to test for AQP abs? cell-based assay according to Franciotta et al.
Franciotta D, Gastaldi M, Sala A, Andreetta F, Rinaldi E, Ruggieri M, Leante R, Costa G, Biagioli T, Massacesi L, Bazzigaluppi E, Fazio R, Mariotto S, Ferrari S, Galloni E, Perini F, Zardini E, Zuliani L, Zoccarato M, Giometto B, Bertolotto A. Diagnostics of the neuromyelitis optica spectrum disorders (NMOSD). Neurol Sci. 2017 Oct;38(Suppl 2):231-236. doi: 10.1007/s10072-017-3027-1. PMID: 29030768.
-- was she sufficiently B cell depleted when experiencing the relapse while on RTX? Yes, still B-cell in the blood test.
-- please report results of brain MRI and follow-up spinal cord MRI
Yes
- Discussion:
-- some of the topical literature on Covid 19 and NMOSD needs to be discussed, see for example
Yes, added
https://pubmed.ncbi.nlm.nih.gov/34446434/
Apostolos-Pereira SL, Campos Ferreira L, Boaventura M, de Carvalho Sousa NA, Joca Martins G, d'Almeida JA, Pitombeira M, Silvestre Mendes L, Fukuda T, Souza Cabeça HL, Chaves Rocha L, Santos de Oliveira B, Vieira Stella CR, Lobato de Oliveira EM, de Souza Amorim L, Ferrari de Castro A, Pereira Gomes Neto A, Diogo Silva G, Bueno L, de Morais Machado M, Castello Dias-Carneiro R, Maciel Dias R, Porto Moreira A, Piccolo A, Kuntz Grzesiuk A, Muniz A, Diniz Disserol C, Ferreira Vasconcelos C, Kaimen-Maciel D, Sisterolli Diniz D, Comini-Frota E, Coronetti Rocha F, Cruz Dos Santos GA, Dadalti Fragoso Y, Sciascia do Olival G, Ruocco HH, Siqueira HH, Sato HK, Figueiredo JA Jr, Cortoni Calia L, Teixeira Dourado ME Jr, Scolari L, Ribeiro Soares Neto H, Melges L, Magno Gonçalves MV, Vellutini Pimentel ML, de Castro Ribeiro M, Gurrola Arambula O, Diniz da Gama P, Leite Menon R, Barbosa Thomaz R, de Rizo Morales R, Sobreira S, Machado SN, Gonsalves Jubé Ribeiro T, Coelho Santa Rita Pereira V, Maia Costa V, da Nóbrega Junior AW, Vieira Alves-Leon S, Mamprim de Morais Perin M, Donadi E, Adoni T, Gomes S, Brito Ferreira M, Callegaro D, Mendes MF, Brum D, von Glehn F; Neuroimmunology Brazilian Study Group. Clinical Features of COVID-19 on Patients With Neuromyelitis Optica Spectrum Disorders. Neurol Neuroimmunol Neuroinflamm. 2021 Aug 26;8(6):e1060. doi: 10.1212/NXI.0000000000001060. PMID: 34446434; PMCID: PMC8404206.
https://pubmed.ncbi.nlm.nih.gov/34429342/
Newsome SD, Cross AH, Fox RJ, Halper J, Kanellis P, Bebo B, Li D, Cutter GR, Rammohan KW, Salter A. COVID-19 in Patients With Neuromyelitis Optica Spectrum Disorders and Myelin Oligodendrocyte Glycoprotein Antibody Disease in North America: From the COViMS Registry. Neurol Neuroimmunol Neuroinflamm. 2021 Aug 24;8(5):e1057.

Reviewer 3 Report
This is an interested case report that confirms the safety and effectiveness of tocilizumab in a patient with NMOSD AQP4-positive and infected by Covid-19, who was not responding to Rituxan.
Although several works suggested that tocilizumab may be a treatment option in NMOSD AQP4-positive patients, further follow-up studies with larger patient samples are needed to assess itseffectiveness.
Please consider a few additional lines in the Discussion related to what is known (or not known) about the role of tocilizumab in APQ4 negative patients, and in patients with relapsing MOGAD.
Author Response
Thank you for the suggestions. Changes are highlighted in violet to add role of tocilizumab in seronegative and MOGAD patients.
Round 2
Reviewer 1 Report
Authors have provided an improved version of the manuscript. However, the manuscript should be English professionally edited to be published in this journal.